

# Minimal Zeeman field requirement
# for a topological transition in superconductors

**Kim Pöyhönen[1,2*], Daniel Varjas[1,2,3], Michael Wimmer[1,2] and Anton R. Akhmerov[1]**

**1** QuTech, Delft University of Technology, P.O. Box 4056, 2600 GA Delft, The Netherlands
**2** Kavli Institute of Nanoscience, Delft University of Technology,
P.O. Box 4056, 2600 GA Delft, The Netherlands
**3** Department of Physics, Stockholm University, AlbaNova University Center,
106 91 Stockholm, Sweden

★ kkpoyhon@gmail.com

## Abstract

Platforms for creating Majorana quasiparticles rely on superconductivity and breaking of time-reversal symmetry. By studying continuous deformations to known trivial states, we find that the relationship between superconducting pairing and time reversal breaking imposes rigorous bounds on the topology of the system. Applying these bounds to $s$-wave systems with a Zeeman field, we conclude that a topological phase transition requires that the Zeeman energy at least locally exceed the superconducting pairing by the energy gap of the full Hamiltonian. Our results are independent of the geometry and dimensionality of the system.

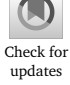

# 1 Introduction

Topological superconductors combine a gapped superconducting bulk with protected zero energy states, with 1D systems supporting Majorana bound states being a particularly interesting example because of their potential use in quantum computation [1]. Most naturally occurring superconductors are topologically trivial, and they have a fully developed gap with an *s*-wave order parameter. A part of the research of topological superconductors therefore focuses on engineering hybrid devices that alter conventional superconductors to drive them through a topological phase transition [2–7].

A recent experiment [8] reported zero bias peaks (a possible signature of Majorana bound states) in a device combining a semiconducting nanowire, a superconducting aluminium shell, and an EuS ferromagnetic insulator, as can be seen in Fig. 1(a). Based on several device characterizations, the authors argue that their data is consistent with EuS inducing a large Zeeman splitting inside the aluminium, but having a negligible direct effect on the semiconducting nanowire. Furthermore, the authors observe that the aluminium shell retains a superconducting gap, and therefore it is likely that the induced Zeeman field is smaller than the superconducting pairing $\Delta$. Combining these observations, the authors conjecture that the system is well modelled by a semiconducting nanowire proximited by a Zeeman-split superconductor, sketched in Fig. 1(b).

This device is therefore very different from prior work [9–13], where the Zeeman splitting is induced by an external magnetic field, and is therefore maximal in the semiconductor due to its large gyromagnetic factor. This experimental observation motivates a study into what constraints apply to the Zeeman field and superconducting pairing in hybrid devices. Several recent works [14–16] have been unable to repeat the observation in similar models, or provided reasoning for why it is unlikely to occur within reasonable parameter ranges [17].

Because the exact material parameters of the experimental device are unknown, we consider the general problem of a time-reversal invariant normal Hamiltonian, combined with a general time-reversal breaking term, and a general superconducting pairing term. We find that the eigenvalues of the combined time-reversal breaking and superconducting term restrict the topological phase of the system: a topological phase transition requires negative eigenvalues to exceed the magnitude of the system gap. We then apply this to the case of a position-dependent Zeeman field and local *s*-wave superconductivity. We find that in absence of other mechanisms of time-reversal symmetry breaking—such as superconducting phase difference or orbital magnetic field—the Zeeman field of a topologically nontrivial hybrid system must at least locally exceed the *s*-wave pairing by the energy gap of the system. This is the case

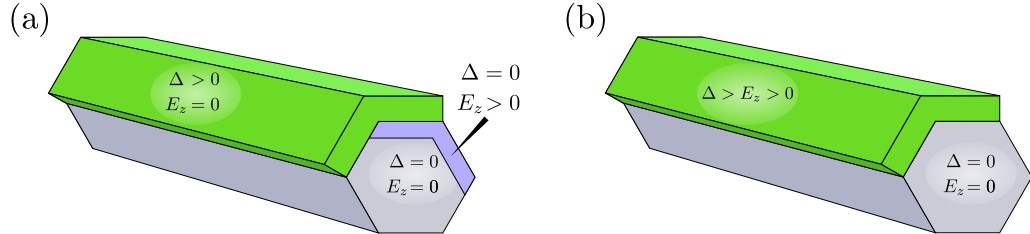

Figure 1: (a) A schematic representation of the experimental setup in Ref. [8]. The system is a heterostructure consisting of three parts: a semiconducting nanowire, an *s*-wave superconductor as well as a magnetic insulator. (b) Effective model of the same, a semiconducting nanowire coupled to an *s*-wave superconductor which features a Zeeman term.

in experimental setups where a magnetic field is applied to a proximitized nanowire, with the Zeeman energy exceeding the proximity-induced pairing; however, if a Zeeman term is instead added to the $s$-wave superconductor, to preserve superconductivity it must generally be smaller than $\Delta$, and such insufficient to effect a topological phase transition.

This article is structured as follows. We first present the general form of the Hamiltonian we are interested in studying. In the following section, we prove a number of conditions on the Hamiltonian which are necessary but not sufficient to achieve nontrivial topologies. We then examine how these apply in the experimentally most relevant case of $s$-wave heterostructures with Zeeman energy terms. Finally, we summarize the results and discuss potential future application.

## 2 Model

The starting point of our investigation is the Bogoliubov-de Gennes Hamiltonian written in the Nambu basis:

$$H_{\text{BdG}} = \begin{pmatrix} H_{\text{N}} & \Delta \\ \Delta^{\dagger} & -\mathcal{T} H_{\text{N}} \mathcal{T}^{-1} \end{pmatrix}, \tag{1}$$

where $H_{\text{N}}$ is the electron Hamiltonian, $\Delta$ is the superconducting pairing, and $\mathcal{T}$ is the time-reversal symmetry operator. Because the superconducting pairing satisfies the constraint $\mathcal{T}\Delta\mathcal{T}^{-1} = \Delta^{\dagger}$, $H$ has particle-hole symmetry with the operator $\mathcal{P} = i\tau_y\mathcal{T}$ (with $\tau_i$ the Pauli matrices in particle-hole space); for details, see Appendix A. We decompose the Hamiltonian into its Hermitian blocks:

$$H = \begin{pmatrix} H_0 + H_M & \Delta' - i\Delta'' \\ \Delta' + i\Delta'' & -H_0 + H_M \end{pmatrix}. \tag{2}$$

Here $H_0$ and $H_M$ are the time-reversal symmetric and time-reversal breaking parts of the electron Hamiltonian, while $\Delta'$ and $\Delta''$ are the Hermitian and anti-Hermitian parts of the superconducting pairing. The separation of $\Delta$ into $\Delta'$ and $\Delta''$ depends on the gauge choice, and therefore a careful choice of gauge is useful in imposing stricter bounds on the topology of $H$, as detailed in Appendix B. We allow the matrices $H_0$ $H_M$, $\Delta'$ and $\Delta''$ to contain arbitrary extra degrees of freedom such as orbital, spin, or position. We will only assume that they are Hermitian and periodic in position space, in order to identify whether the Hamiltonian (2) may have a gap closing at the energy $E = 0$—a necessary condition of a topological transition [18]. Because in the thermodynamic limit the gap of a large system with periodic boundary conditions equals to the band gap of the dispersion relation, this allows us to draw conclusions about topological transitions in any dimensionality and with arbitrary geometry.

## 3 Trivial topology from bounds on the energy gap

Our intuition derives from the Anderson's theorems [19], which established that in $s$-wave superconductors, disorder in $H_0$ does not close the energy gap due to the time reversal symmetry. We expect that for $s$-wave-like $\Delta'$, the gap must be closed by terms breaking time reversal, and that a system is trivial if the pairing dominates over such terms. This intuition is confirmed by our random matrix theory simulation, which we share as an interactive notebook at this URL for the readers' benefit. Making the natural generalization to the systems with an arbitrary spatial structure of the pairing and the Hamiltonian, we first consider the case where $\Delta'$ is positive definite, and $H_M$ is sufficiently small.

**Theorem 1.** *The energy gap of H is bounded from below by the smallest eigenvalues of each of* $\Delta' \pm H_M$. *If this bound is greater than 0, H is topologically trivial.*

*Proof.* Let the smallest eigenvalues of $\Delta' \pm H_M$ be $\lambda_\pm$. Assume that $\lambda_\pm > 0$ and that $H$ has an eigenvalue $E$ with $\lambda_+ > E > -\lambda_-$, and therefore

$$\det \begin{pmatrix} H_0 + H_M - E & \Delta' - i\Delta'' \\ \Delta' + i\Delta'' & -H_0 + H_M - E \end{pmatrix} = 0. \tag{3}$$

We apply the Hadamard transformation $U = \frac{1}{\sqrt{2}}(\tau_x + \tau_z)$ to get

$$\det \begin{pmatrix} \Delta' + H_M - E & H_0 + i\Delta'' \\ H_0 - i\Delta'' & -\Delta' + H_M - E \end{pmatrix} = 0. \tag{4}$$

By our assumption, $\Delta' + H_M - E$ is positive definite and hence invertible. Equation (4) is therefore equivalent to

$$\det \left[ \Delta' + H_M - E \right] \det \left[ -\Delta' + H_M - E - (H_0 - i\Delta'')(\Delta' + H_M - E)^{-1}(H_0 + i\Delta'') \right] = 0. \tag{5}$$

Introducing matrices $X = \Delta' - H_M + E$, $Y = (\Delta' + H_M - E)^{-1}$, and $Z = H_0 + i\Delta''$ (by assumption $X$ and $Y$ are both positive definite), we rewrite this equation as

$$\det X \det \left[ X + Z^\dagger Y Z \right] = 0. \tag{6}$$

Since

$$v^\dagger \left[ X + Z^\dagger Y Z \right] v = v^\dagger X v + (Zv)^\dagger Y (Zv) \geq v^\dagger X v > 0, \tag{7}$$

for an arbitrary complex vector $v$, the left hand side of the equation above is a product of two determinants of positive definite matrices, and we arrive to a contradiction. Therefore $\lambda_+ > E > -\lambda_-$ is impossible, or after applying the particle-hole symmetry $|E| \geq \max(\lambda_+, \lambda_-)$.

To show that $H$ is topologically trivial, we note that due to the form of the lower bound proved above, we can smoothly deform $H_0 \to 1, \Delta'' \to 0$ without closing the gap. Further, because $\Delta'$ and $\Delta' \pm H_M$ are positive definite, so are $\Delta' \pm tH_M$ with $0 \leq t \leq 1$, and we can likewise deform $H_M \to 0$. Hence $H$ is topologically equivalent to $\tau_z + \Delta'\tau_x$ with a positive definite $\Delta'$, which can be further deformed to a trivial onsite potential by letting $\Delta' \to 0$ again without closing the gap. □

Theorem 1 excludes a positive semi-definite $\Delta'$, and therefore does not directly apply to most experimental setups where parts of the system lack superconducting pairing. In order to apply our reasoning to such systems, we assume that the resulting hybrid Hamiltonian has a finite energy gap $E_G$ and observe that creating such a gap requires a finite perturbation. This allows us to formulate the following:

**Theorem 2.** *Both $\Delta' \pm H_M$ must have eigenvalues smaller than or equal to $-E_G$, with $E_G$ the gap of $H$, for $H$ to be topological.*

*Proof.* Let $\lambda_\pm$ again be the smallest eigenvalues of $\Delta' \pm H_M$ respectively, and $\lambda_0 = \max(\lambda_+, \lambda_-) > -E_G$. The case of $\lambda_0 > 0$ is covered by Theorem 1, so we consider $\lambda_0 \leq 0$. Define $\Delta_0 = \frac{1}{2}(E_G - \lambda_0) > 0$, so that $E_G > \Delta_0 > -\lambda_0$. We define $H'(t) = H + t\Delta_0\tau_x$. If $E_i$ are the eigenvalues of $H$, and $E_i'(t)$ are the eigenvalues of $H'(t)$, by Weyl's inequality they satisfy

$$E_i - t\Delta_0 \leq E_i' \leq E_i + t\Delta_0. \tag{8}$$

For this reason, for the gap $E_G'(t)$ of $H'(t)$ is

$$E_G'(t) \geq E_G - t\Delta_0 > 0. \tag{9}$$

Hence, $H$ is topologically equivalent to

$$H'(t=1) = \begin{pmatrix} H_0 + H_M & \Delta_1 - i\Delta'' \\ \Delta_1 + i\Delta'' & -H_0 + H_M \end{pmatrix}, \tag{10}$$

with $\Delta_1 = \Delta' + \Delta_0$. However, for this Hamiltonian, we have $\Delta_1 \pm H_M = \Delta' \pm H_M + \Delta_0$, with smallest eigenvalues $\lambda_\pm + \Delta_0 \geq \lambda_0 + \Delta_0 > 0$. By our previous results, $H'(t=1)$ and hence $H$ is then trivial. $\qquad\square$

Finally, to highlight the application of Theorem 2 to time-reversal invariant systems, we restate it as follows.

**Theorem 3.** *If $H$ has an energy gap $E_G$, and $H_M = 0$, then $H$ is topologically trivial unless $\Delta'$ has eigenvalues smaller than $-E_G$.*

In other words, the superconducting pairing of time-reversal invariant topological superconductors must have negative eigenvalues with magnitude exceeding the size of the gap. This is an improvement of the result of Ref. [20], in which the authors showed that a class DIII superconductor in one or two dimensions cannot be topological if $\Delta'$ is positive semidefinite.

# 4 Application to $s$-wave systems

We now apply our findings to the heterostructures with proximity-induced $s$-wave pairing, where time-reversal symmetry is broken by a Zeeman splitting.

Because $\Delta'$ and $H_M$ are now both local in space, we immediately obtain the required eigenvalues by comparing matrix elements. Let the local magnitude of the $s$-wave pairing on site $i$ be $\Delta_i$ and the local magnitude of the Zeeman energy be $B_i$, so that the eigenvalues of $\Delta' \pm H_M$ are $\Delta_i \pm B_i$. Then, applying Theorems 1 and 2, we find

1. If $\forall i : \Delta_i - B_i > 0$, then the system is topologically trivial, and the energy gap is bounded from below by $\min_i(\Delta_i - B_i)$.

2. If the system has an energy gap $E_G$, then the system must be topologically trivial if $\max_i(B_i - \Delta_i) < E_G$.

The second finding also establishes an upper bound of $E_G \leq \max_i(B_i - \Delta_i)$ for the gap of such a system in the topological phase.

Let us consider how this translates to the effective model proposed in Ref. [8], displayed in Fig. 1(b), where a Zeeman splitting induced in the superconductor. It is seen that by Theorem 2, any system of this type cannot enter a topologically nontrivial phase, as $\Delta \geq E_z$ everywhere. Indeed, any model where the Zeeman splitting induced in the semiconductor is small compared to other energy scales, even if nonvanishing, will not work; in order to induce a topological phase transition, it must have a magnitude larger than the system gap. In the actual experimental setup, as seen in Fig. 1(a), the EuS region does have $E_z \gg \Delta$, and as such the theorems presented here do not necessarily prohibit nontrivial phases when the coupling between and InAs cannot be neglected. For a more in-depth analysis of how a ferromagnetic insulator can induce topological superconductivity we refer the reader to the recent works. Specifically, Refs. [14–16] offer an alternative explanation, where the proximity effect between the InAs nanowire and the EuS magnetic insulator is enhanced by the presence of the Al superconductor, while Ref. [17] proposes a spin-dependent tunnelling that essentially enhances the time-reversal symmetry-breaking term.

# 5 Conclusion

We established that time-reversal symmetry breaking must exceed the superconducting pairing by at least the spectral gap in order for a superconductor to be topological. Our results offer a quick way to rule out the appearance of Majorana particles in hybrid devices in the parameter regimes where Zeeman field is sufficiently weak.

This bound is most relevant in *s*-wave systems with a pure Zeeman field. Where a Zeeman field is induced directly in the superconductor, but not in the semiconductor, such as in the model of Ref. [8], we find that topological superconductivity is prohibited. In systems containing a magnetic flux comparable with a single flux quantum, the norm of the time-reversal symmetry-breaking parts of the Hamiltonian becomes large, therefore making our bound easily exceeded. On the other hand, by instead assuming preserved time-reversal symmetry, our theorems also extend a previous result [20], that time-reversal invariant conventional superconductors in one or two dimensions are trivial, to arbitrary dimension.

Because our method does not assume anything about the orbital structure of the superconducting gap, it applies also beyond *s*-wave systems, e.g. in Refs. [21, 22], where several different pairings, some of which are positive definite, are considered. Thus our results may be useful in future research where similar types of unconventional superconductivity are relevant.

**Author contributions**   A.R. A. formulated the initial project goal based on a random matrix theory simulation, and helped streamline the proofs. K. P. derived the analytical results with input from M. W., and wrote the original draft of the paper. D. V. improved the clarity of the proofs. All authors discussed the results and contributed to writing the manuscript.

**Funding information**   This work was supported by the NWO VIDI Grants 680-47-537 and 016.Vidi.189.180, a subsidy for top consortia for knowledge and innovation (TKI toeslag), by Microsoft Quantum, and by the Swedish Research Council (VR) as well as the Knut and Alice Wallenberg Foundation.

# A   Applications beyond the Nambu basis

In the main text we made the assumption that the Hamiltonian has the typical Nambu form with particle-hole symmetry $\mathcal{P} = i\tau_y\mathcal{T}$, with $\mathcal{T}$ not acting in particle-hole space. However, this assumption is stronger than necessary; only some of the properties this assumption yields are needed. Additionally, in many cases the Hamiltonian will be in a basis or symmetry class that necessitates a different form of the symmetry operators. Hence, in this Appendix we will discuss to what extent the theorems can be applied to these cases.

By examining the proofs, we see that it is not necessary for the Hamiltonian to be in the Nambu basis, nor for the symmetry operators to take any particular form. Indeed, our results can be applied to any Hamiltonian of the general form of Eq. (2), if it features either particle-hole or chiral symmetry, and if adding a unit matrix to $\Delta'$ without breaking the discrete symmetries preserving the topology of the Hamiltonian. We will outline the reasoning below.

First, we note that any Hermitian matrix of size $2N \times 2N$ can be written in the form of Eq. (2). The requirement for particle-hole or chiral symmetry stems from using the property that topological phase changes coincide with gap closings at zero energy.

The third assumption requires more elaboration. The reason for the requirement is that Theorems 2 and 3 rely on adding such a term. It could be argued that Theorem 1 does not, and that said step in the other two could be circumvented by adding another positive definite

matrix instead. However, below we will show that any Altland-Zirnbauer symmetries that permit a positive definite matrices $\Delta'$ must also be compatible with $\Delta' = rI$ where $I$ is the identity matrix and $r \in \mathbb{R}$. The proof of Theorem 1 also involves a step in which $H_0$ is deformed to unity and hence requires $\tau_z$ to be allowed by any symmetries, but this is not necessary in a general sense; deformation to another trivial Hamiltonian works equally well.

Finally, we note that in the Nambu basis assumed in the main text, it is easily seen that $\tau_x$ does not break the discrete symmetries.

**Theorem 4.** *Any discrete Altland-Zirnbauer-type symmetry that permits a nonvanishing positive or negative definite block matrix $\Delta'$ will also be compatible with one that is proportional to the identity.*

*Proof.* First, we note that this is trivially true for chiral symmetries: $U\Delta' = -\Delta'U$ necessitates that the eigenvalues of $\Delta'$ be symmetric around zero. Hence, we focus on antiunitary symmetries.

Let $\tau$ be a real Hermitian unitary matrix and $\Delta$ be a Hermitian matrix operating in another subspace, so that we can write $\tau \otimes \Delta$ as $\Delta\tau$. Assume there exists a unitary matrix $U$ so that

$$U^\dagger \Delta^* \tau U = s\Delta\tau, \tag{11}$$

where $s = \pm 1$. Clearly $s = -1$ corresponds to particle-hole symmetry, and $s = +1$ corresponds to time reversal. Define the matrices $a = U - s\tau U\tau$ and $b = U + s\tau U\tau$, so that $U = a + b$, and note that $a\tau = -s\tau a$ and $b\tau = s\tau b$.

We can obtain the following relations for $a$ and $b$:

$$ab^\dagger = -ba^\dagger \qquad aa^\dagger + bb^\dagger = 1. \tag{12}$$

Further, by substitution of $U = a + b$ into Eq. (11), we see that

$$b^\dagger \Delta^* b - a^\dagger \Delta^* a = \Delta \tag{13}$$

$$a^\dagger \Delta^* b = b^\dagger \Delta^* a. \tag{14}$$

Because $\Delta$ is a Hermitian matrix, its eigenvectors satisfy the equation

$$\Delta|\lambda\rangle = \lambda|\lambda\rangle, \tag{15}$$

for real eigenvalues $\lambda$. Let us assume that $a \neq 0$, i.e, the particle-hole (time reversal) unitary matrix has a component that commutes (anticommutes) with $\tau$. Applying Eq. (13), multiplying by $a$ from the left, and then making use of the other relations described above, we find that we must have

$$\Delta[a|\lambda\rangle]^* = -\lambda[a|\lambda\rangle]^*. \tag{16}$$

This can only be true $\forall|\lambda\rangle$ if either $a\Delta = 0$ or if $\Delta$ has eigenvalues symmetric around zero. Hence, $\Delta$ cannot be positive (or negative) definite unless $a = 0$. By substituting $\tau = \tau_x$ or $\tau = \tau_z$, we see that if the symmetries prohibit identity matrices on the block (off-) diagonal, all other positive or negative definite matrices are likewise forbidden. $\qquad\square$

## B  On the gauge dependence of $\Delta''$

As noted in the main text, the distinction between $\Delta'$ and $\Delta''$ is essentially one of gauge, and hence the theorems, relying on $M_\pm \equiv \Delta' \pm H_M$, are gauge dependent as stated. However, it is easy to see if that a unitary transformation $U = \cos\frac{\theta}{2}\tau_x + \sin\frac{\theta}{2}\tau_y$ provides a new

$\Delta'_\theta = \Delta' \cos\theta + \Delta'' \sin\theta$. for which we can repeat the steps of each proof using $M_\pm(\theta) = \Delta'_\theta \pm H_M$. In order for the system to be topological, it must meet the criteria for each $\theta$ separately; as such, for a final, gauge independent formulation we can consider $\max_\theta[\max(\lambda_+(\theta), \lambda_-(\theta))]$ with $\lambda_\pm(\theta)$ being the smallest eigenvalue of $M_\pm(\theta)$. The proofs in the main text are still valid; this formulation can at most extend the region proved to be trivial. As usual, care must be taken that the range of $\theta$ considered only covers terms allowed by the discrete symmetries of the system; for the standard symmetries $\mathcal{T} = i\sigma_y K$ and $\mathcal{P} = \tau_y \sigma_y K$, this does not pose additional restrictions.

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
