# Peer review of "Minimal Zeeman field requirement for a topological transition in superconductors"

_SciPost Physics, doi:SciPost Phys. 10, 108 (2021)_

## Round 2 · Referee Report · Anonymous (Referee 1) · 2020-12-16

Strengths

  • motivation by recent experiments that are not well understood
  • clear presentation of the model and its mathematical properties
  • thorough discussion of the different steps of the derivation

Weaknesses

  • the main physics message is a bit hidden (behind the mathematical theorems)
  • the main novelty of the work is not clearly emphasized

Report

From my point of view, the main message of the paper is that a finite contribution to a gapped Hamiltonian is needed to close the energy gap and generate a topological phase transition. This message is not new.

It might be formulated in a more rigorous way in the present manuscript than in previous works. However, from a physics point of view, it is a known statement. Therefore, I personally do not agree that the scientific discovery of the authors fulfill one of the demanding acceptance criteria of SciPost Physics.

Requested changes

I suggest to emphasize more the novelty of the work. A small sentence along these lines is written after Theorem 3 but not in the Introduction (if I have not missed it).

  • validity: top
  • significance: good
  • originality: good
  • clarity: high
  • formatting: excellent
  • grammar: perfect

Author:  Kim Pöyhönen  on 2021-03-02  [id 1277]

(in reply to Report 1 on 2020-12-16)

We thank the referee for their comments on the manuscript.
Their overall concern appears to be that the physical message and novelty of the
work have not been sufficiently emphasized, as listed in both the weaknesses
as well as the suggested changes parts of the report. In order to address this,
we have elaborated on the physical motivation in both the introduction and in
section 4. We hope that these changes improve the clarity of our presentation
and help to illustrate the physics involved.

We also wish to comment on the claim that "From my point of view, the main message of the
paper is that a finite contribution to a gapped Hamiltonian is needed to close
the energy gap and generate a topological phase transition." This is an unfortunate
misunderstanding of the main message of our manuscript, which we hope to clarify both in
our answer and the amended version.

The problem we set out to answer was whether a spin-split conventional superconductor
coupled to a nanowire, as in Ref. 8, can be topologically nontrivial. As discussed in
Section 4, we found that he answer is categorically no regardless of dimension or geometry.
This was previously known for a simple model, but not in general, as evidenced not only by
the fact such a model was used in Ref. 8, but also by the several papers that have since been
written about it, e.g arXiv: 2011.06547, 2011.06567, 2012.12934, and 2012.00055.
Our manuscript differs from these other recent works in a few respects; typically, they consider
a geometry similar to Ref. 8 and suggest alternative explanations for the observations; our
manuscript, in turn, makes no assumptions about the geometry and instead categorically shows
that the original approach cannot work. As such, our work is cited in several of the above
(2012.00055, 2011.06567, and 2012.12934) as evidence for why an alternative explanation is needed.
Further, our results apply to a wide range of systems.
The report highlights a few sentences at the end of Section 3, where we mention that the third
theorem essentially generalizes Ref. 20 to arbitrary dimension; this was included as an
example of how one might apply the theorems outside of the initial problem statement.
We also note that Theorem 2 (of which Theorem 3 is a corollary) is an even more general case,
applying to systems with broken time reversal symmetry.

In summary, our work provides mathematical tools that can be applied to a wide range of systems.
While our main results, formulated in theorems 1 and 2, appear abstract and general, the application
of these to the setup of Ref. 8, which even the experts working in the field have overlooked,
demonstrates their usefulness.

---

## Round 2 · Referee Report · Anonymous (Referee 2) · 2021-1-2

Strengths

  • Formulation in terms of a set of theorems, with clearly formulated conditions and otherwise general validity.

  • Relevance to recent experiments on Majorana bound states.

Weaknesses

  • One may argue that, as with most mathematical theorems, strictly speaking the conditions for the theorems derived in the manuscript do not apply to realistic physical systems. But I think that this issue is well addressed in the article's introduction.

Report

This articles proves a general bound on when topological superconductivity can occur in a system without orbital magnetic field, inspired by the recent experiments of the Copenhagen group that show evidence for the existence of topological superconductivity in a semiconducting nanowire proximity coupled to both a superconductor and a magnetic insulator (Vaitiekenas et al., Ref. 8). In the standard theoretical description, the effect of the magnetic insulator is to induce an effective exchange field in the superconductor S and in the semiconductor N. The manuscript proves that inducing the exchange field in S is not a viable mechanism to drive the N-S hybrid into a superconducting state. This statement, which is formulated as a theorem, is that for topological superconductivity to exist, the Zeeman splitting must locally exceed the s-wave superconducting gap at that position. For the experiment, this implies that the direct exchange coupling between the magnetic insulator and N is essential for the appearance of topological superconductivity.

The no-go theorem derived in this manuscript is consistent with a number of other recent theory articles (all appearing on the arXiv within one month of submission of the present manuscript) that also address the origin of topological superconductivity in the devices studied by the Copenhagen group. What sets the present manuscript apart from the other articles on the subject, is that it makes a statement of general validity.

The formulation of the conditions for the theorem and the presentation of the mathematical steps is very clear, and the physical conclusions and their range of applicability are discussed in a transparent manner. The manuscript is not longer than necessary. Its introduction is to the point. I have no reservations recommending this manuscript for publication in scipost physics.

Requested changes

No changes are necessary.

  • validity: top
  • significance: high
  • originality: high
  • clarity: high
  • formatting: excellent
  • grammar: excellent

Author:  Kim Pöyhönen  on 2021-03-02  [id 1278]

(in reply to Report 2 on 2021-01-02)

We thank the referee for their comments and for recommending the manuscript for publication in SciPost Physics.

---

## Round 3 · Author Response

Dear editor,
We have reviewed and addressed the referee evaluations. We thank both referees for their feedback. The first referee considers the main results of our work not new. This is an unfortunate misunderstanding of our paper, as also seen from the report by the second referee and the references to our work in the recent literature. In the resubmitted version we have rewritten the introduction to clearer present the physics problem we have addressed, and we have clearly stated the new results. We expect that together with our response to the referee, this will be sufficient to change the referee's opinion.
We have reviewed and addressed the referee evaluations. We thank both referees for their feedback. The first referee considers the main results of our work not new. This is an unfortunate misunderstanding of our paper, as also seen from the report by the second referee and the references to our work in the recent literature. In the resubmitted version we have rewritten the introduction to clearer present the physics problem we have addressed, and we have clearly stated the new results. We expect that together with our response to the referee, this will be sufficient to change the referee's opinion.

---

## Round 3 · List of Changes

Extended the introduction and section 4 to clarify physical motivation; included a schematic figure. Expanded the Conclusions section to better detail some of the implications of the manuscript.

---

## Editorial Decision

published